# Collagen pre-strain discontinuity at the bone—Cartilage interface

**Waqas Badar**[1], **Husna Ali**[1], **Olivia N. Brooker**[1], **Elis Newham**[1], **Tim Snow**[2], **Nicholas J. Terrill**[2], **Gianluca Tozzi**[3], **Peter Fratzl**[4], **Martin M. Knight**[1], **Himadri S. Gupta**[1] *

**1** Institute of Bioengineering and School of Engineering and Material Science, Queen Mary University of London, London, United Kingdom, **2** Harwell Science and Innovation Campus, Diamond Light Source, Harwell, Didcot, United Kingdom, **3** School of Engineering, University of Greenwich, Chatham Maritime ME4 4TB, UK, **4** Department of Biomaterials, Max-Planck-Institute of Colloids and Interfaces, Potsdam Wissenschaftspark, Golm, Germany

* h.gupta@qmul.ac.uk

**Data Availability Statement:** All data files are available from the Queen Mary Research Online (QMRO) repository (https://doi.org/10.17636/10179773).

## Abstract

The bone-cartilage unit (BCU) is a universal feature in diarthrodial joints, which is mechanically-graded and subjected to shear and compressive strains. Changes in the BCU have been linked to osteoarthritis (OA) progression. Here we report existence of a physiological internal strain gradient (pre-strain) across the BCU at the ultrastructural scale of the extracellular matrix (ECM) constituents, specifically the collagen fibril. We use X-ray scattering that probes changes in the axial periodicity of fibril-level D-stagger of tropocollagen molecules in the matrix fibrils, as a measure of microscopic pre-strain. We find that mineralized collagen nanofibrils in the calcified plate are in tensile pre-strain relative to the underlying trabecular bone. This behaviour contrasts with the previously accepted notion that fibrillar pre-strain (or D-stagger) in collagenous tissues always reduces with mineralization, via reduced hydration and associated swelling pressure. Within the calcified part of the BCU, a finer-scale gradient in pre-strain (0.6% increase over ~50μm) is observed. The increased fibrillar pre-strain is linked to prior research reporting large tissue-level residual strains under compression. The findings may have biomechanical adaptive significance: higher in-built molecular level resilience/damage resistance to physiological compression, and disruption of the molecular-level pre-strains during remodelling of the bone-cartilage interface may be potential factors in osteoarthritis-based degeneration.

## Introduction

The BCU plays a crucial biomechanical role in enabling pain-free articulation and smooth transmission of shear and compressive stresses across diarthrodial joints [1]. Structural breakdown and alterations in bone-cartilage cellular communication at the bone-cartilage interface has been proposed as a key early-stage marker for OA [2]. OA is a debilitating musculoskeletal degenerative condition, with over 300 million people affected [3] and a global prevalence over 20% for individuals over 40 [4]. Development of structural biomarkers for early-stage degeneration in interfaces like the bone-cartilage interface would enable early diagnosis and better

**Funding:** The study was funded by the following: Engineering and Physical Sciences Research Council, EP/V011235/1, Dr Himadri Shikhar Gupta Engineering and Physical Sciences Research Council, EP/V011383/1, Dr. Nicholas J Terrill Biotechnology and Biological Sciences Research Council, BB/R003610/1, Dr. Nicholas J Terrill, Dr Himadri Shikhar Gupta Medical Research Council, MR/R025673/1, Dr Himadri Shikhar Gupta Diamond Light Source, SM25602-2, Dr Himadri Shikhar Gupta, Dr. Nicholas J Terrill, Mr. Waqas Badar, Dr. Tim Snow, Prof Martin M. Knight.

**Competing interests:** The authors have declared that no competing interests exist.

therapies [5]. As these changes will originate at the molecular- and supramolecular level, understanding the functional design of such interfaces at the nanoscale in both healthy and degenerative conditions is of importance.

The bone-cartilage interface, like other biological interfaces [6], exhibits gradients in both structure and mechanical properties, resulting in gradients in strain during physiological activity. On one side, there is a layer of articular cartilage (AC) with depth-dependent gradients in fibre orientation (first described by the Benninghof architecture [7]), type II collagen/proteoglycan content ratios, and hydration [8], and (more recently identified) nanoscale collagen fibrillar pre-strain and intramolecular disorder [9]. As shown in Fig 1, collagen fibrils in the AC traverse a tidemark into a layer of calcified cartilage (CC) [10, 11], enabling a firm anchorage of the AC cartilage to the underlying subchondral bone (SCB). Below the SCB is the more open spongy or trabecular bone (TB) network. Scanning X-ray scattering and electron microscopy measurements on the calcified mineral phase in the calcified plate (CP) (the combined

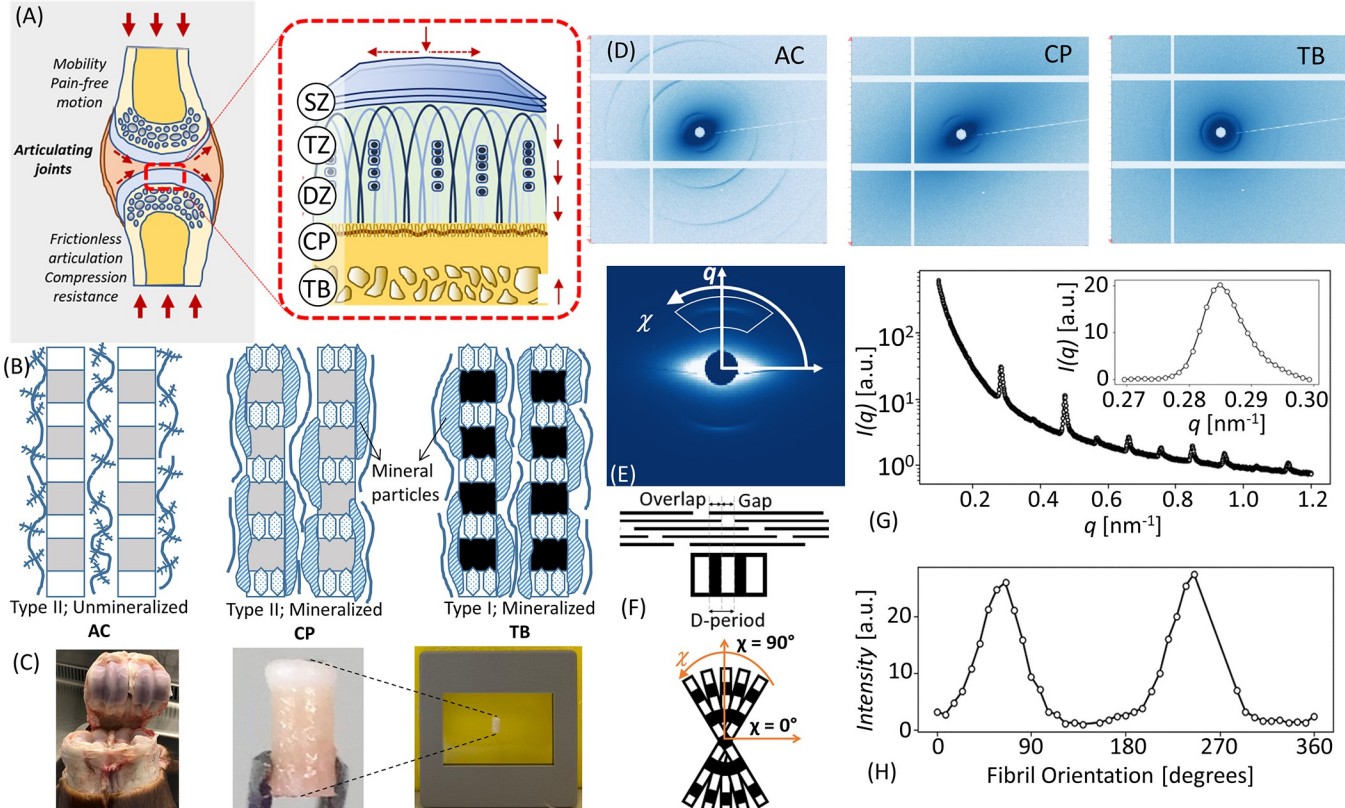

**Fig 1. Bovine BCU explants and small angle X-ray scattering (SAXS) data analysis.** (A) Schematic of joint structure showing the collagen fibril orientation from parallel to the articulating surface in the superficial zone, to vertical in the deep zone and calcified plate. Dashed arrows indicate, schematically, the directions of shear (horizontal) and compressive (vertical) stresses. *Abbreviations for region classification*: SZ: superficial zone; TZ: transitional zone; DZ: deep zone: CP: calcified plate, TB: trabecular bone (B) Fibrillar microstructure in AC, CP, and TB zones. Type II collagen fibrils are found in AC and calcified cartilage (CC), with mineral particles as well in CC, whereas type I collagen fibrils are found in SCB and TB. (C) Bone-cartilage unit (BCU) cores of 5mm length and 2mm diameter extracted from bovine metacarpophalangeal joints and placed in cassette holder for SAXS scanning while kept hydrated. (D) 2D SAXS patterns from AC, CP, and TB. Low diffuse SAXS with clear peaks and oriented ellipse in AC, high diffuse SAXS with average peaks and oriented ellipse in CP and high diffuse SAXS with average peaks and no orientation for ellipse in TB. (E) Azimuthal and radial integration to get the I(q) and I(χ) profiles to infer nano structural parameters (D-period, fibrillar orientation and degree of alignment). (F) Fibril schematic: Collagen D-period, characterised by the repeating gap-overlap regions causes by the staggered fibril stacking and fibrillar orientation. (G) I(q) plot, showing peaks due to collagen D-period, particularly clear at the 3[rd] and 5[th] order after azimuthally integration (0–360˚) around the diffraction pattern for the q-radial range 0.1–1.2nm[-1]. Right top plot is 3[rd] order collagen peak after background subtraction. (H) I(χ) profiles after corrected intensity with Gaussian model fitting to the peak. Gaussian fitting was used to infer parameters (fibril orientation and $\rho$).

calcified cartilage and subchondral bone), have revealed gradients in mineral composition [12], crystallinity, and size [10], and recently, differences in the mineral crystallite thickness between healthy and osteoarthritic patients [11, 13]. The gradient in compressive properties from cartilage (1-10MPa) to subchondral bone (GPa) reduces high interfacial shear stresses which could otherwise lead to cracks and delamination between the softer cartilage and the stiffer bone. Even with graded mechanical properties, physiological loading still induces high apparent strains of up to 5% in the calcified cartilage interface, measured using digital volume correlation (DVC) [14]. Such a combination of high *in vivo* loading along with gradients across mechanically dissimilar materials requires a structurally optimized design at the material level, to prevent microcrack propagation [15] and enable energy absorption during loading.

At this material level, the ECM of the bone-cartilage interface in articular cartilage, the calcified plate and trabecular bone is essentially a collagen fibrillar composite, with a variable composition of the extrafibrillar phase (Fig 1): glycosaminoglycan (GAG)/proteoglycan (PG) rich in articular cartilage, while predominantly mineralized (carbonated apatite) in the calcified plate and trabecular bone [16, 17]. Biochemically, articular cartilage and calcified cartilage contain predominantly Type II collagen versus Type I collagen in the subchondral and trabecular bone. Both fibril types exhibit a characteristic D-banding periodicity (D~65–67 nm) which is detectable with methods such as small-angle X-ray scattering (SAXS) [18, 19] and atomic force microscopy [20]. Shifts in the D-period have been linked to fibril strain in bone, tendon, and cartilage [18, 21–23], although recent work highlights the need to consider intrafibril tilt as well, in tissues like cornea [22, 24].

Internal strains and stresses in the fibril phase (both at equilibrium and under loading) are expected to be crucial for structural integrity, joint function and mechanobiological stimuli. The fibril composite in cartilage can be considered as a pre-stressed hydrogel, where swelling pressure exerted by the hydrated GAG/PG is restrained by the collagen fibres leading to tensile pre-strain on the collagen fibrils, identified by an increase in D-period. Recently, our synchrotron X-ray scattering analysis of the ECM nanostructure in the articular cartilage has shown that this fibrillar pre-strain exhibits an increase from the superficial/transitional to the deep zone (adjacent to the calcified plate) by ~1% (corresponding to ~5 MPa) [9]. Such nanoscale gradients are associated with the well-established micro- and macro-level gradients in fibril orientation from parallel to perpendicular to the joint surface known previously [25]. Under compressive loading or synthetic ageing, we demonstrated that these fibrillar gradients in pre-strain are disrupted, indicating the gradients play a mechanically important role in the stabilisation of articular cartilage [9]. However, a limitation of the previous study [9] was that the mechanically crucial calcified plate and underlying bone was not considered, as biopsies were taken by cutting the tissue at the bone-cartilage interface, in common with many other studies measuring the mechanical properties of cartilage explants *ex-vivo*. Indeed, mineralized collagen (as found in the calcified plate and trabecular bone) has usually a lower D-period than unmineralized (as in articular cartilage), which has been explained via loss of water inside the fibril [26–28]. The pre-strain thus increases from the transitional/superficial to deep zone cartilage [9], lying adjacent to the underlying mineralized tissue with a putatively lower D-period. The calcified plate is subject to significant tissue-level and microscale strains in loading [14]. However, the corresponding ultrastructural D-period variation in the calcified plate (relative to the articular cartilage and trabecular bone) has not been measured.

This present study aims to measure the gradient in pre-strain in collagen D-period and related ultrastructural properties in the bone-cartilage interface, focusing on the calcified plate, to better characterise this biomechanical interface critical for healthy joint function. A bovine model of the bone-cartilage interface from metacarpophalangeal (MCP) joints is used [9, 18].

High brilliance synchrotron X-ray SAXS mapping, in the fast fly-scan modes, allows relatively rapid data acquisition across micro- and macroscale tissue regions. The high X-ray flux at synchrotrons enable localized measurements with a micro-beam, necessary to detect micron-scale variation of nanoscale ECM parameters. We carry out SAXS mapping with multiple (three) spatial resolutions 1) across the entire articular cartilage, calcified plate, and trabecular bone, 2) focused on the articular cartilage and calcified plate, and 3) a high-resolution mapping of the calcified plate region. Our results will identify whether potentially mechanically important equilibrium or pre-strain gradients exist at the ECM level, helping establish baselines for deviations in OA and other musculoskeletal degeneration.

## Materials and methods

### Sample preparation

BCU explants were extracted from a MCP joint (Fig 1C), of freshly slaughtered adult bovine steers (16–24 months of age), delivered by the local abattoir (C Humphreys & Sons- Blixes Farm Shop, Chelmsford, UK). Bovine cartilage was selected for use in the experiment, as it is both widely available, has less variability as compared to human cartilage samples and is the next best biomechanical approximation of human cartilage [29]. The intact joints were washed with biological detergent followed by immersion in disinfectant (Chemgene) for 15 minutes (this procedure did not affect the internal cartilage and bone tissue as the joints were intact). Joints were then opened using sterile disposable scalpel blades (Swann Morton) in a biohazard safety cabinet. The metacarpal bone was removed, and two metacarpal condyles were isolated from each joint using a high-speed saw. Extracting intact bone-cartilage cores required a custom sample-preparation process; a method inspired by the work of Aspden and co-workers was used [30]. First, each condyle was placed in a custom-designed 3D-printed holder with an axis of rotation matching that of the condyle flexion axis. A bench drill (Axminster), with a diamond coated coring drill bit was used to produce cores of 2mm diameter, under constant irrigation. The condyle was rotated around the sample-holder axis so that, at each extraction point on the condyle, the drill-face was incident locally normal to the condylar surface; this ensured flat cylindrical samples. The drill was used to core to a 5 mm-depth at five locations on the proximal surface of each condyle. To complete the extraction, the intact condyle was removed from the 3D-printed holder, and the reverse face of the condylar section was abraded continuously with a high-speed Dremel rotary tool equipped with a burring-bit, until each core detached. Both drilling systems were used with constant phosphate buffered saline solution (PBS) (Sigma-Aldrich, Poole, UK) irrigation, with the drilling advancing slowly to minimize heat and mechanical damage. Finally, the extracted cores were cut to an equal length of 5mm by removing excess TB, using a Buehler IsoMet low-speed saw and a 3D printed sample holder under constant irrigation. Final cores, as shown in Fig 1C, were placed in Eppendorf tubes with Dulbecco's Modified Eagle's Medium–low glucose (DMEM) (Sigma-Aldrich, Poole, UK), and stored at -20˚C until the experiments were conducted at Diamond Light Source (DLS).

### SAXS measurement protocols

SAXS scanning of the BCU were performed at beamline I22 [31], Diamond Light Source (DLS, Harwell Science and Innovation Campus, Didcot, UK). The beam size at the sample was measured to be 20µm, beam energy of 14keV and the sample to detector distance at 5.8m. The samples were mounted in film cassette windows (Fig 1C), which were sealed using a bilayer Kapton film arrangement with enclosed PBS, to ensure hydration over the duration of the SAXS scan. SAXS scanning was performed with a Pilatus 2M detector [32] (pixel size 172 µm;

resolution 1475 × 1679 pixels) with an exposure time of 1s for each SAXS measurement. SAXS scans were carried out in "fly-scan" mode where the sample-stage moves continuously during the measurement; this ensures rapid data collection whilst still maintaining spatial registration. Three different SAXS scan settings were used:

*Regular scan*: For n = 6 samples, a rectangular 2D area scan of 2.6mm (vertical ($y$)) × 0.8mm (horizontal ($x$)) was performed with 40 μm increments in both $x$- and $y$-direction across the BCU, which included the full AC and CP tissue but only a part (1.36mm) of the ~4 mm long TB region.

*Detailed scan*: For a single sample, to obtained detailed SAXS data at the bone-cartilage interface region, a square 2D area scan of 0.4mmx0.4mm at the interface was performed with 5μm increments in the $x$ and $y$-direction.

*Full-length scan*: For a single sample, to obtain the SAXS data for the full-scale length of the extracted BCU core shown in Fig 1C (extending from AC through the full length of TB), a rectangular 2D area scan of 5mm (vertical ($y$); distance from joint surface) × 0.38mm (horizontal ($x$); distance parallel to joint surface) was performed, with 20 μm increments in both $x$ and $y$-direction.

## Data processing

SAXS analysis was carried out via a combination of DAWN (www.dawnsci.org), an Eclipse-based software framework developed at Diamond Light Source [33], with customized model fitting routines (in Python; Anaconda Python distribution from Continuum) applied to the 1D integrated intensity files generated by DAWN. Fibril-level ultrastructural parameters (fibrillar D-period, fibrillar orientation and degree of fibrillar alignment) were extracted from SAXS patterns, via both prior published protocols developed in our group [9, 18], and by use of new (non-parametric) methods of intensity peak centre and width estimation.

## DAWN data reduction

The 2D-rastered maps of 2D SAXS data (1 per array point) for each scan (shown in Fig 1D), were reduced to 2D arrays of 1D intensity profiles. The radial $I(q)$ and azimuthal (angular) $I(\chi)$ files were extracted as a function of scattering vector ($q$) or azimuthal angle ($\chi$) depending on integration mode (azimuthal and radial, respectively) shown in Fig 1E, and saved as text files. The *Processing* perspective in DAWN was used for the batch reduction of the data files [33]. As well as a mask file specific for the Pilatus detector, which excluded the dead-regions (seen as white strips in Fig 1D) between charge-coupled device (CCD) plates or the beam-stop.

## I(q) processing for Fibril D-period

Fig 1G shows an $I(q)$ plot from our data with DAWN showing the radial meridional peaks, particularly clear at the 3rd and 5th order diffraction peaks (wavevector range 0.1 to 1.2 nm$^{-1}$). For collagen-specific analysis, the 3rd order meridional peak was used because it is clearly visible in both (uncalcified) cartilage, the calcified plate and in trabecular bone (in contrast to the 5th order peak which is clearly visible in articular cartilage [9] but less clear in calcified tissue). After azimuthal integration on DAWN, the $I(q)$ data sets produced for each scan point were read by custom Python scripts to perform the peak analysis for the desired parameters, as described below:

1. *Total SAXS intensity*: The total SAXS intensity was calculated from the area under the $I(q)$ curve. Total SAXS intensity, primarily from diffuse scattering, is related to the amount of interfacial area between nanoscale inclusions like mineral platelets and surrounding matrix

in bone [34], or between fibrils and the extrafibrillar matrix. Here, we use it to a) distinguish between the bone-cartilage core and surrounding fluid and b) distinguish between the calcified and uncalcified tissue regions in the scan, as calcified tissue regions have much higher diffuse SAXS intensity arising from the mineral phase.

2. *Collagen peak intensity*: The collagen peak intensity is defined as the total area under the 3rd order meridional collagen peak after diffuse background SAXS subtraction. Diffuse SAXS background subtraction is done as per our previous protocols [9, 35], by a linear interpolation of the diffuse signal from the left ($q$ = 0.27 nm$^{-1}$) to the right ($q$ = 0.3 nm$^{-1}$) of the 3rd order meridional peak. The collagen peak intensity is related to a) the total amount of collagen fibrils/unit volume present in the scattering volume b) the degree of intrafibrillar order [18] and c) the orientation of the fibril in 3D space [36].

3. *Fibril D-period*: The fibril D-period arises from the axial periodicity in electron density along the collagen fibril due to the gap-overlap stacking of tropocollagen molecules inside the fibril [37]. Shifts in the D-period can be linked to axial fibril strain (e.g. [18, 38–40]), and in cartilage, changes in the D-period have been interpreted as changes in tensile pre-strain due to variations in swelling and osmotic pressures in the proteoglycan-rich extrafibrillar phase [9, 18]. For all points in the 2D scan, we set a threshold of 0.025 [arbitrary units] in collagen peak intensity ((from 2) above), above which the D-period (and associated peak parameters) was estimated. The need for the threshold is because estimating the fibril D-period is only meaningful where there are sufficient collagen fibrils at a point in the scattering volume satisfying the Ewald diffraction condition and generating a clear 3rd order meridional peak.

To estimate D-period, initially a parametric peak fitting approach (starting with Gaussians) was used, following prior work in our group [9, 41] and using the *lmfit* nonlinear least squares fitting package in Python [42]. However, we found that there was a pronounced rightward asymmetry in the peak profile, which was not visible in prior work [9, 41]. The possible structural reasons for this effect (arising from fibre diffraction), and its implications for the measured D-period will be explored and justified in detail in the **Discussion**. Here, we note only that the peak asymmetry is visible due to the recent upgrade of X-ray optics at the I22 beamline (in 2019) which led to much higher resolution in the peak profile which could not be resolved in earlier work [9, 18]. Due to the asymmetric peak nature, we use a nonparametric method-of-moments approach, treating the $I(q)$ profiles (with the diffuse linear background intensity subtracted) as continuous distributions around the central wave vector (q) values, to find the parameters centre($\mu$) and sigma($\sigma$). The first moment ($q_{FM}$ = $\mu$), that is the expectation value $\mu$, gave the position of the peak ($q_0$ in our prior notation [9]) and the standard deviation gave $\sigma$ (the width of the peak). The following equations were used to calculate the first moment, the second moment and the standard deviation.

$$\text{First moment, } q_0 \text{ or } q_{FM} = \frac{\sum q I_b(q)}{\sum I_b(q)} \qquad \text{Eq 1}$$

$$\text{Second moment, } q^2_{SM} = \frac{\sum q^2 I_b(q)}{\sum I_b(q)} \qquad \text{Eq 2}$$

$$\text{Standard deviation, } \sigma = \sqrt{q^2_{SM} - q_{FM}{}^2} \qquad \text{Eq 3}$$

where $q$ and $I_b(q)$ are the wave vector and (background corrected) radial intensity values around the 3rd order peak, as shown in Fig 1G. From $q_0$ the D-period was calculated as (where

n is the peak order, n = 3 for the 3$^{rd}$ order meridional peak):

$$D = n\frac{2\pi}{q_0} \qquad\qquad Eq\ 4$$

To represent the SAXS scans as 2D images, parameters (like D-period, total SAXS intensity and collagen peak intensity) were displayed as a colour-scale bit-map, where the colour value was related to the value of the calculated parameters through a colour-map.

## I($\chi$) processing for fibril orientation and degree of fibrillar alignment

1D azimuthal intensity profiles I($\chi$) at each scan point were generated to obtain parameters characterising the angular collagen fibril distribution–the average fibril orientation $\chi_0$, and degree of fibrillar alignment ($\rho$: random order$\rightarrow\rho$ = 0, uniaxial alignment $\rightarrow$ $\rho$ large). In a similar manner to the *I(q)* analysis, the background diffuse scattering needs to be corrected for. A three-ring subtraction method was used to obtain corrected intensity value, by radially averaging the intensity just outside of the peak and subtracting this average from the radial average on the peak, as described previously by us [43]:

$$Collagen\ Intensity = Peak\ intensity - \S(inner\ intensity + outer\ intensity) \qquad Eq\ 5$$

It is noted that across the full azimuthal 0-360° range, some parts of the detector were blocked due to dead-regions between Si pixel modules or the beam-stop. A restricted range from 0 to 180$^o$ where such blockages were minimized was therefore used. This restricted angular range provides the full information due to the theoretical $\chi\rightarrow\chi$+180° symmetry of the scattering for SAXS (from to the Ewald sphere being approximated as a plane). The corrected intensity peaks, I$_{corrected}$ (after background subtraction in **Eq** 5) were fitted with a Gaussian model with zero baseline using a custom Python program using the *lmfit* nonlinear fitting package [42], as shown in Fig 1H and following our prior work [9, 43, 44].

The $\rho$-parameter was calculated from I$_{corrected}$ data by evaluating the Gaussian model fit results, using the following equation

$$\rho = (I_{corrected\_max} - I_{corrected\_min})/I_{corrected\_mean} \qquad Eq\ 6$$

where I$_{corrected\_max}$ and I$_{corrected\_min}$ are the maximum and minimum values in the fit for I$_{corrected}$. Note this parameter (for unmineralized and mineralized collagen) is quantitatively different from the Rho-parameter used earlier by Fratzl and colleagues [45] for mineralized tissues, (though qualitatively both are measures of the degree of alignment of the fibril) and is hence denoted with the symbol $\rho$.

A custom Python script using the *matplotlib* package [46] was used to display the colour map of the calculated orientation parameters, where the angle of the line shows the fibrillar orientation. Both the length of the line, and the colour value of each pixel corresponds to the degree of fibrillar alignment ($\rho$), similar to prior work [10]. The definition of degree of fibrillar alignment $\rho$ used here differs from the rho-parameter used earlier for degree of mineral particle alignment in bone [45], but has qualitative similarities in that both are used to represent angular anisotropy of nanoparticle/nanofibre alignment and are close to zero for randomly oriented nanoscale inclusions.

*Zonal assignment*. To analyse differences in these SAXS parameters across tissue zones, we first used scalar measures from the SAXS pattern to classify a point as belonging to 1) articular cartilage, 2) calcified plate, or trabecular bone. Within articular cartilage, we further subcategorized points as arising from the superficial, transitional, or deep zone. First, a threshold of 2.0 [arbitrary units] in total SAXS intensity was used to differentiate between articular cartilage

(low total SAXS intensity) and calcified plate (high total SAXS intensity due to mineral scattering). The dense cortical calcified plate and the spongy trabecular bone regions were differentiated based on the vertical $y$-coordinate, as the cylindrical sample geometry leads to a simplified 1D (vertical $y$) variation in the sample tissue structure, and a visually clear demarcation between the calcified plate and the trabecular bone. For zones within in the articular cartilage region, the *degree of fibrillar alignment ρ* and *direction of fibril orientation χ0* were used as differentiators: points with low $\rho$ ($<$ 2.1) with randomly aligned fibrils were assigned to the transitional zone (TZ). Points above the transitional zone and with $\chi_0$ around 0˚ or 180˚ (horizontal) was assigned to the superficial zone (SZ), and points below with $\chi_0$ around 90˚ were assigned to the deep zone (DZ). The justification for this assignment is the well-known arcade-like Benninghof structure [7] for the fibril orientation in articular cartilage. For all samples, a 5-color map (SZ/TZ/DZ/CP/TB) of the scans was generated, based on these assignments, and visually inspected to check that this classification did not result in any artefacts, e.g. an assignment of a point deep inside the trabecular bone to the calcified plate category; all 5-color maps for regions are also shown in S1 Fig.

## Data analysis, fitting and statistical analysis

To compare the SAXS-derived parameters between tissue zones, two parallel approaches were used: 1) a *zonally-averaged approach* across-samples and 2) an *intra-sample* approach.

For 1), the mean value of each SAXS-parameter was calculated for each zone and each sample ($n$ = 6 samples). To account for the inter-sample variability, we used a paired $t$-test (two-tailed) protocol to compare the SAXS-parameters between adjacent zones (SZ-TZ, TZ-DZ, DZ-CP and CP-TB). In the complementary approach 2), for each sample-scan, every SAXS pattern in the scan was assigned to zones SZ, TZ, DZ, CP or TB based on the protocol described above, and reduced to obtain the relevant SAXS parameters. Statistical comparison of each SAXS-parameter was carried out by 1-way ANOVA, with the zone as the variable. After Anova, Tukey's honestly significant difference (HSD) pair-wise tests were carried out between data in different zones. Statistical test-results were reported on a per-sample basis and histograms of the SAXS parameters were also plotted. The approach 1) of paired $t$-test analysis is reported in the main text, and the intra-sample approach 2) in the Supplementary Data. This is because the main scientific question is whether there are (on average) differences in SAXS-derived ultrastrucutural parameters across zones in the bone-cartilage unit.

The paired t-tests were carried out in Microsoft Excel and the 1-way analysis using R (www.r-project.org). For data fitting, custom Python scripts (Anaconda Individual Edition; www.anaconda.com) using the packages *lmfit* [42] and *matplotlib* [46] were used for the analysis of $I(q)$ and $I(\chi)$ profiles created in DAWN, along with Microsoft Excel.

## Results

### Spatial gradients of fibrillar-parameters across the BCU

In the following, we present quantitative 2D mapping and zonal analysis of ultrastructural SAXS-based parameters across the BCU at different length-scales. Fig 2A shows the classification of the different tissue regions into five zones: in articular cartilage, the superficial (SZ), transitional (TZ) and deep (DZ) zones, and in the calcified tissue, the dense CP (calcified plate) and the trabecular bone (TB). Fig 2B and 2D shows 2D maps of the micro-spatially varying nanostructural parameters derived from $I(q)$ (D-periodicity, total SAXS intensity and collagen peak intensity) across the BCU. The D-period exhibits two clear features: i) a gradient of increasing D-period from the surface of the joint to the deep zone in articular cartilage, which is maintained in the calcified plate, followed by ii) a distinctly lower D-period range across the

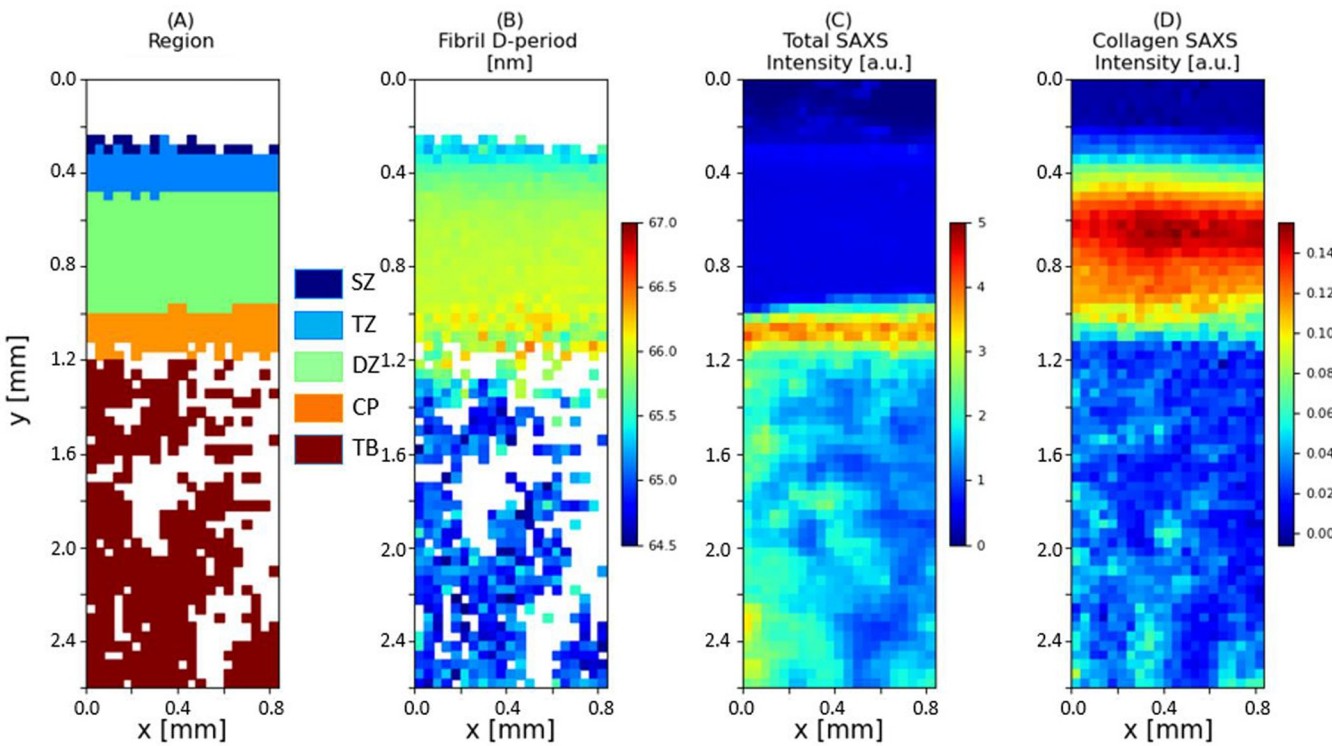

**Fig 2. Variation of fibril pre-strain and ultrastructural SAXS measures across the BCU.** Colour plots of 2D SAXS scan on a bone-cartilage unit (BCU) (A) Region, with the acronyms following the region classification in Fig 1A, (B) Fibrillar D-period (nm), (C) Total SAXS intensity (a.u.) and (D) Total diffraction intensity from the background-corrected meridional collagen peak intensity (a.u.). The plots show the variation in these nano structural parameters across the BCU.

underlying trabecular bone (where no spatial gradient is seen). It is noted that the D-period in the calcified plate is higher than the trabecular bone despite both being comprised of mineralized collagen. Fig 2C also shows that the total SAXS intensity *a)* is low in articular cartilage, *b)* shows a clear high band in the calcified plate, due to strong SAXS scattering contrast from the mineral relative to the surrounding organic phase, and *c)* exhibits a patchy cellular structure in the TB, reflective of the porous mineralized tissue structure in trabecular bone.

In a complementary manner, for the fibril orientation derived from I($\chi$), Fig 3 shows the 2D parameter map for the orientation direction and extent of orientation $\rho$ for the same sample shown in Fig 2. The orientation follows the well-established Benninghof ultrastructure of fibres, with superficial-zone fibrils aligning along the horizontal plane (around 0˚/180˚ angle), no preferred direction in the transitional zone (short lines within each voxel), fibrils in the deep zone and calcified plate aligning vertically (around 90˚ angle) and no preferred direction in trabecular bone (short lines). The superficial zone, characterised by fibrils oriented parallel to the joint surface, is most easily identified by fibre direction coupled with a higher degree of alignment $\rho$ than in the adjacent transitional zone. The colour scale (representing $\rho$) shows higher $\rho$ values in the deep zone and the calcified plate as compared to transitional zone and trabecular bone.

## Significant differences in fibrillar-parameters between zones

To extract the mean values of the SAXS-derived parameters, and estimate statistical differences between zones, first histograms are plotted on a per-zone basis (S2 Fig). Mean values and

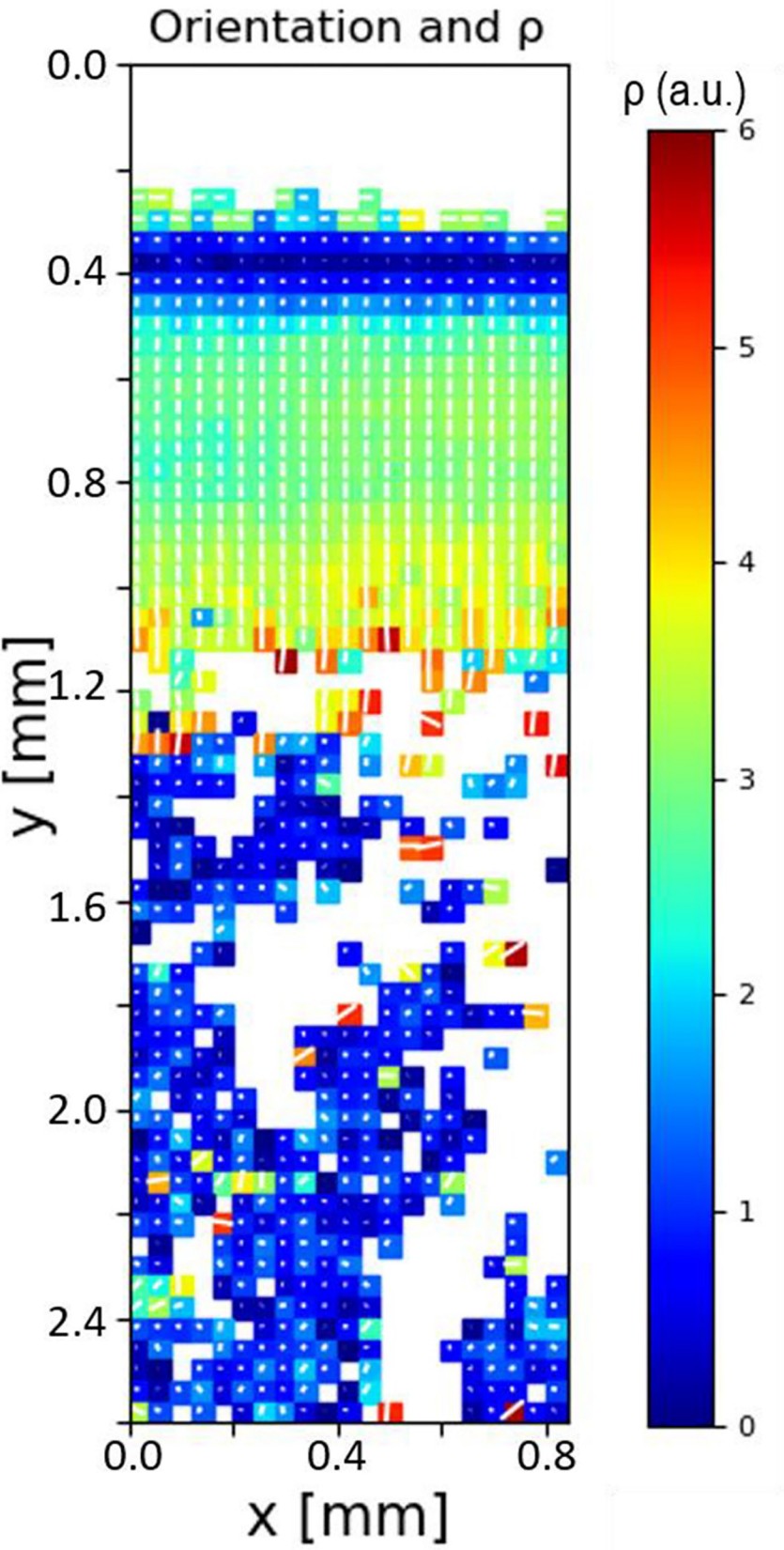

**Fig 3. Fibril orientation and alignment across the BCU.** Colour map (for the sample in Fig 2) for collagen fibril orientation (degrees) and $\rho$ (a.u.) parameter, where the length of the white line and the colour scale corresponds to $\rho$, and the direction of the white line indicates the fibrillar orientation across BCU. There is a clear increase in the degree of fibrillar alignment at the bone-cartilage interface, with the fibrillar orientation perpendicular to articulating surface. However, there is random orientation with lower degree of alignment in TB.

standard deviations are calculated per zone. To compare the differences in mean values, on a per-sample basis, Fig 4 reports the result of paired $t$-tests between adjacent tissue zones. The D-period increases significantly on going from the superficial to the transitional zone, and from the transitional to the deep zone. The differences between the deep zone and the calcified plate are not significant, but a highly significant reduction is observed in D-period between the calcified plate and the trabecular bone. For the degree of orientation $\rho$, significant reductions occur on going between the superficial and transitional zone, and significant increases between the transitional and deep zone. Like the D-period, $\rho$ is not significantly different between the deep zone and calcified plate but is much lower in the trabecular bone versus the calcified plate. As described in *Materials and Methods*, as an alternate analysis method (2), we carried out 1-way ANOVA tests on a per-sample basis (contrasting with the paired approach comparing per-sample means here), giving similar results (**S2 Table**).

## Qualitative differences in meridional SAXS peak profile between zones

Since the zonal variation in D-period is small (e.g., ~65.2 nm to 66.0 nm in Fig 4) as is often the case in collagenous tissues [18, 47], it is a reasonable question as to how significant these changes are, and whether our new non-parametric method of peak estimation may be artefactually influencing the difference. First, we note that the magnitude of the effect (~1%) is comparable to the fibril-level strains experience in bone under deformation to fracture [43]. To further clarify this, we plot $I(q)$ profiles, with and without diffuse SAXS background subtraction, from each of the 5 different zones in an example BCU (Fig 5) to see how evident these peak-profile differences are in the original data itself. It is clear, especially from the background subtracted data (Fig 5B), that there are distinct changes in peak position and shape across the zones; most notably, the right-shifted $I(q)$ for TB demonstrates the lowered D-period seen in the colour-plots in Fig 2 and quantified in Fig 4. Fig 5B also shows that in AC, the total SAXS intensity due to the collagen peak alone (excluding the diffuse SAXS scattering) is maximum in the DZ, and least in the SZ. This effect may arise due to a combination of varying collagen concentration as well as degree of fibrillar ordering (higher ordering corresponds to higher SAXS peak intensity [9, 47]). Interestingly, the collagen peak height is also larger in CP than in the TB region.

## High resolution SAXS mapping at the interface shows finer-scale gradients

To map out the ultrastructural variation of the fibril parameters at the calcified interface itself, Figs 6 and 7 show the high-resolution colour maps with 5 μm spacing. Laterally-averaged line-profiles of the same data are shown in Fig 8, to more clearly shown the spatial variation with distance from the articular cartilage/calcified plate interface. The location at the top of the images corresponds to the transition from unmineralized deep zone articular cartilage to the calcified plate, evidenced by the lower total SAXS intensity coupled with the high collagen peak intensity. As indicated by the arrows in Fig 8A, over a region of about 70 μm, the D-period is observed to increase from ~65.4 nm to 65.8 nm (~0.6% increase) (**dashed line a1**), following which the D-period is approximately constant over ~70 μm (**dashed line a2**). The D-period then exhibits a decrease over ~170 μm to ~65 nm (**dashed line a3**), following which the trend stabilises, which corresponds to reaching the trabecular bone region.

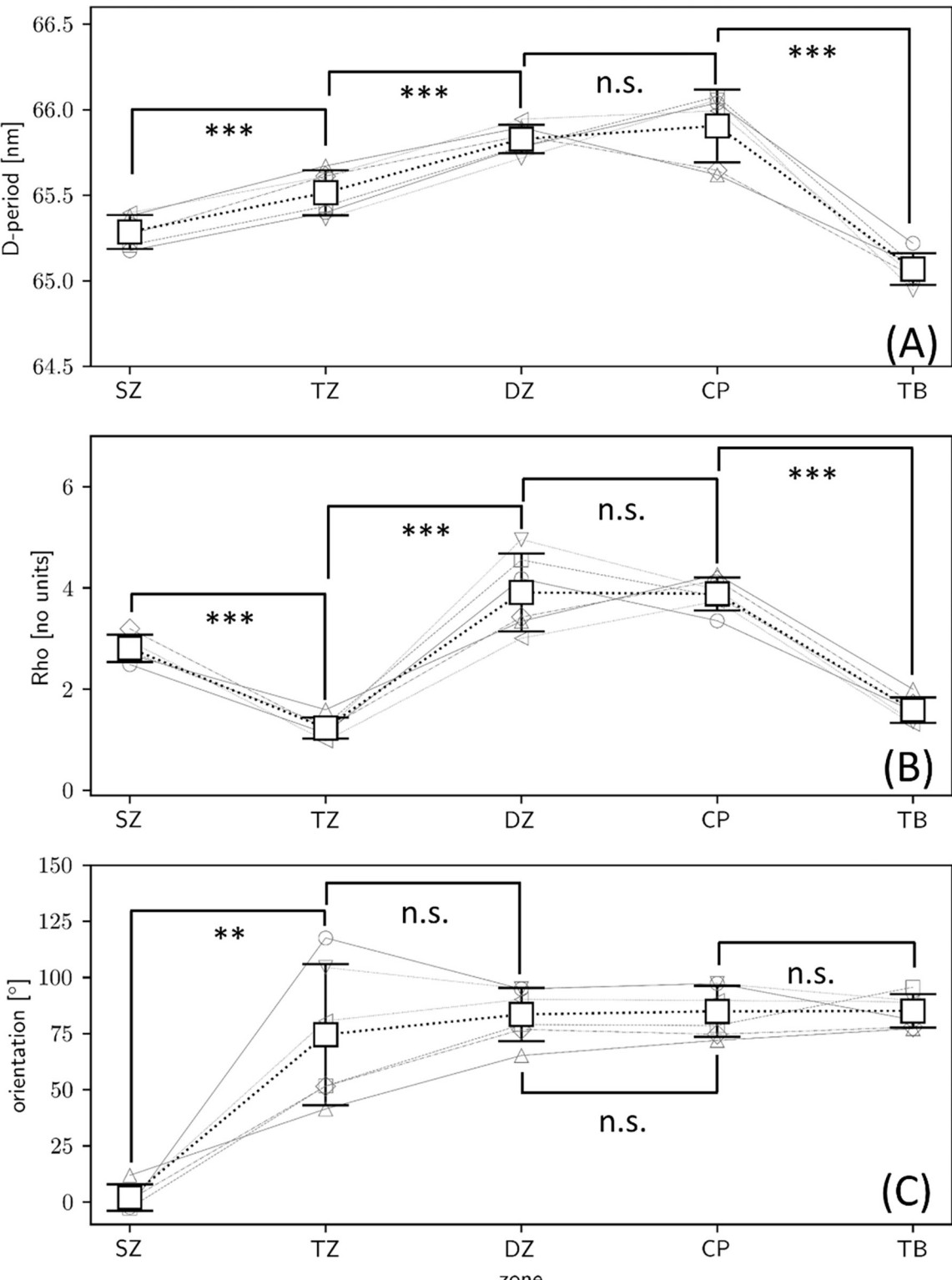

**Fig 4. Significant difference in fibrillar pre-strain, alignment, and orientation across the BCU.** Bar plots of the (A) fibril D-period (pre-strain), (B) degree of alignment ($\rho$) and (C) fibril orientation direction. The SAXS ultrastructural measures are calculated on a per-zone basis for each sample (small symbols, grey outline), which are averaged across the regular scan (n = 6) samples and shown as white squares; error bars: standard deviation across samples.

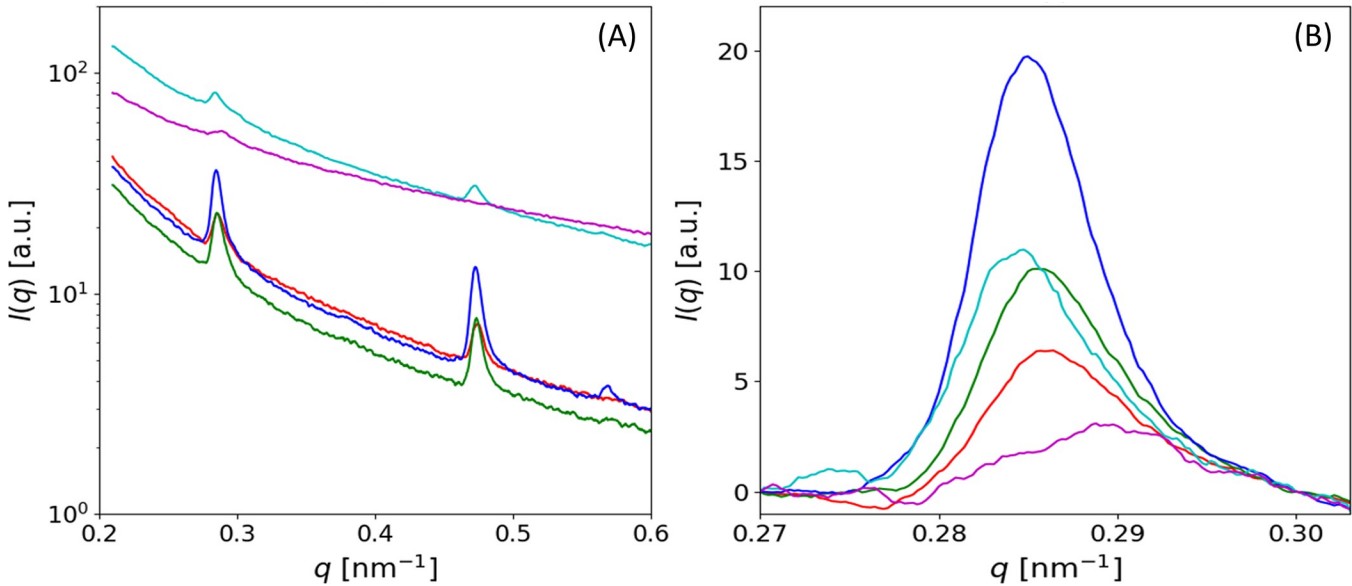

**Fig 5. Variation of SAXS intensity profiles across zones.** (A) I(q) plot, showing 3rd order and 5th order peaks due to collagen D-period, after azimuthally integration around the diffraction pattern for the q-radial range 0.2–0.6nm⁻¹. (Red: SZ, Green: TZ, Blue: DZ, Cyan: CP and Magenta: TB). (B) The 3rd order collagen peaks across the BCU after the diffuse SAXS background subtraction.

Interestingly, on comparing Fig 8A and 8B, the D-period variations show different correlations with the degree of fibrillar alignment ρ, depending on where one looks in the bone-cartilage interface. For the transition from AC to CP (on the left in Fig 8A and 8B), the increase in D-period is accompanied by a decrease in ρ by about 20% (dashed lines a1 and b1). On the right-hand side in Fig 8A and 8B (also indicated by the region "SCB" in Fig 7), there is a transition to lower ρ (dashed line b3), followed by further decline as the TB region is approached (dashed line b4). In contrast to the AC/CP transition behaviour, on the right-hand side (CP/TB transition), the D-period decreases with reduction in ρ (dash lines a3 and b3). On the left-hand side in Fig 8C, orientation of the fibrils is predominantly perpendicular to the AC/CP

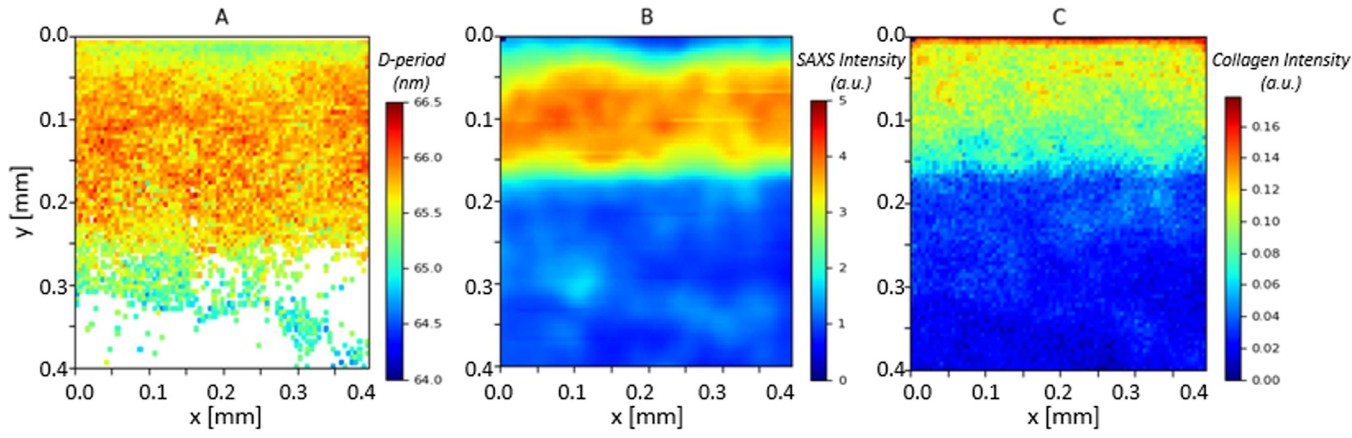

**Fig 6. High spatial resolution map of fibril pre-strain and ultrastructural SAXS measures at the calcified/uncalcified boundary in the BCU.** (A) D-period, (B) SAXS total intensity and (C) SAXS collagen intensity. The uncalcified articular cartilage is at the top of the image (blue shaded region near the ordinate value of 0.0mm for B); the calcified plate (CP) is from ordinal values of 0.0–0.2mm (yellow-red regions in B), and the regions from 0.2mm and below are the trabecular bone (TB) region. In A), the D-period shows a lower average value near the interface. SAXS collagen intensity is higher toward the interface.

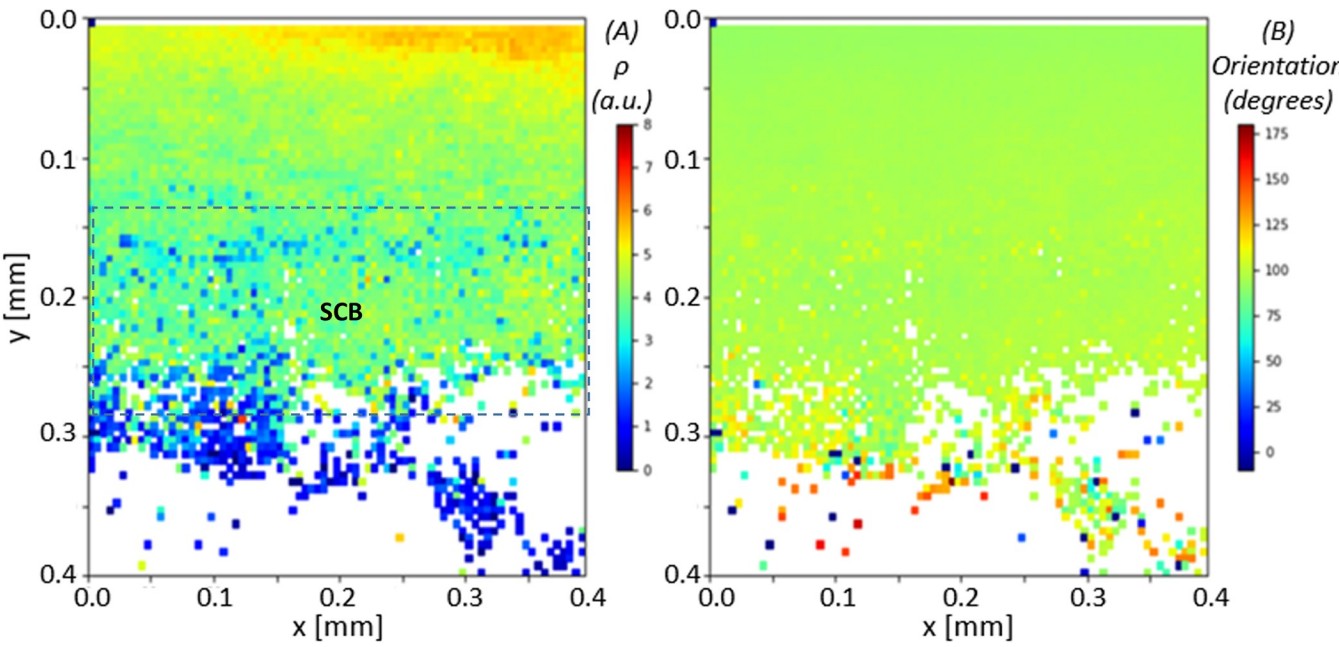

**Fig 7. High spatial resolution map of fibril degree of alignment and orientation at the calcified/uncalcified boundary in the BCU.** (Same sample as in Fig 6) (A) degree of alignment $\rho$ and (B) collagen fibril orientation. From A) the tissue shows a high degree of fibrillar alignment when transitioning from the deep zone of articular cartilage to the calcified plate, and in B) we see fibres predominantly orientated in the angle of 90˚ (green colour), perpendicular to the articulating surface. In the lower part of the calcified plate (dashed rectangle; ordinal values from 0.15mm to 0.3mm) the $\rho$ values are lower than in the upper part; it is suggested this is subchondral bone as opposed to calcified cartilage. From 0.3 mm downward, the degree of orientation is much lower, and orientation is more random, characteristic of trabecular bone.

interface ($\chi_0 = 90$˚), which is a well-known "anchoring" type of architecture for the uncalcified AC to adhere to the SCB. The variance in orientation is less toward AC/CP interface (evidenced by a more homogeneous colour distribution as shown in Fig 7B). On moving toward the TB-region, we see a trend to wider ranges of fibril orientations $\chi_0$, coupled with overall lower degree of alignment $\rho$, which is low in TB.

## Discussion

In summary, the main new findings in this paper are:

- Existence of a variation in fibrillar D-period (linked to fibril pre-strain), extending across the uncalcified articular cartilage (shown by us before [9]), which stabilizes between the deep zone and the underlying calcified plate, i.e. across the bone-cartilage interface.

- A reduction of the D-period in the underlying trabecular bone to the values characteristic for Type I mineralized collagen.

- A finer-scale modulation of this gradient at the deep-zone/calcified plate interface, with high-resolution scans showing a short-length scale (~80 μm) drop and increase in D-period.

- Highly significant differences in D-period and in the degree of alignment $\rho$ across all zones except between the adjacent deep zone and calcified plate.

- Evidence of two distinct regions with altered nanostructure inside the CP, from both fibrillar alignment and pre-strain.

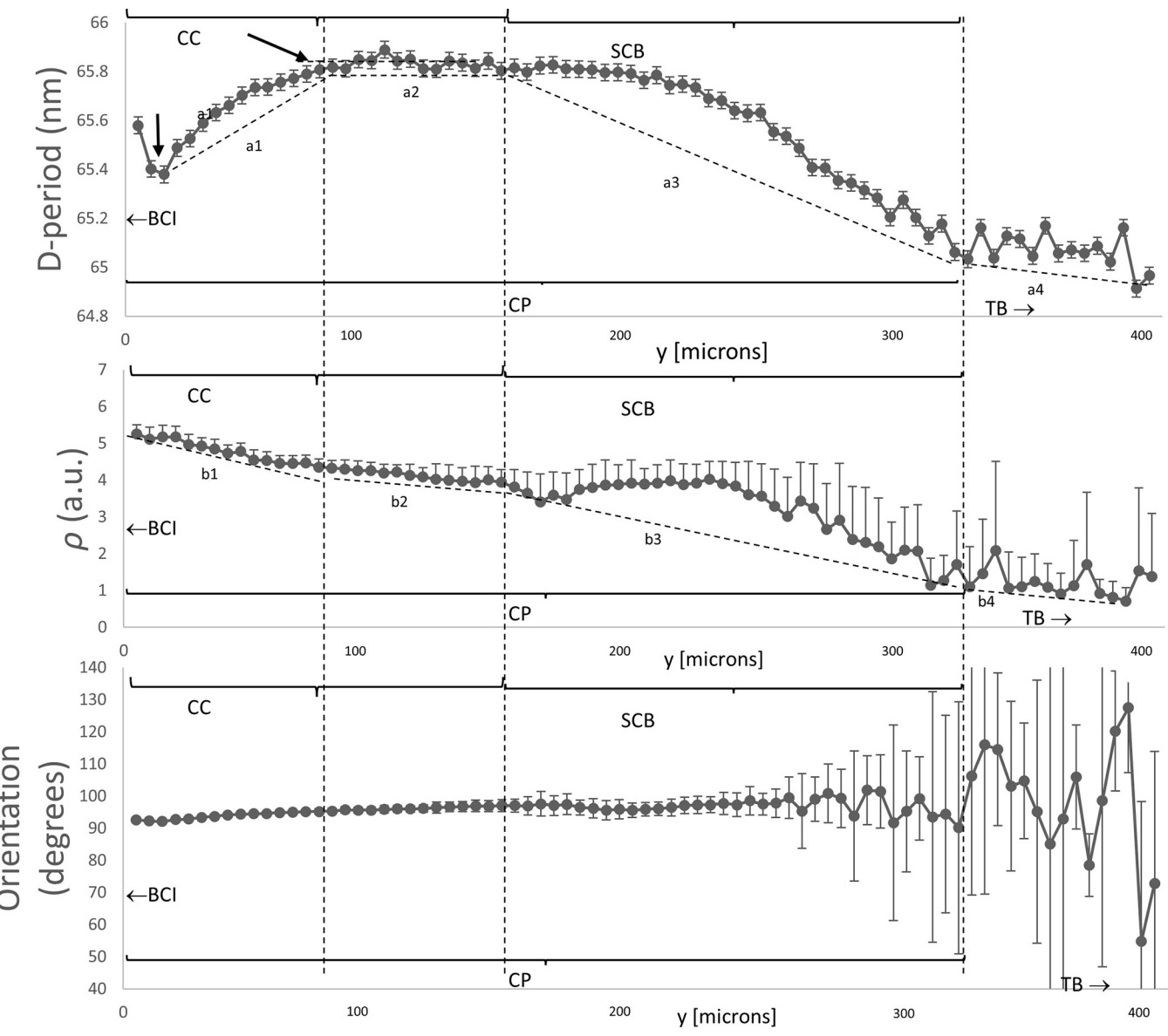

**Fig 8. Variation of fibril D-period, degree of orientation and direction of orientation across the bone-cartilage interface.** 1D line plots, averaged in the horizontal direction, for the high resolution 2D scans in Figs 6 and 7. (A) The mean D-period (B) mean $\rho$ and (C) mean orientation, averaged for each y-coordinate, displayed with error bars representing standard error of the mean. Horizontal (abscissa)-axis is in microns. Interface between calcified and uncalcified cartilage is on the left, indicated by bone-cartilage interface with left-arrow. In A), it is noted that D-period rises from a local minimum toward the interface between the calcified plate and articular cartilage (on the left); it stabilises at the first vertical dashed line ~100 μm to the right of the interface. The right two vertical dashed lines indicate a proposed demarcation of the calcified plate (CP) into calcified cartilage (CC) and subchondral bone (SCB), at the same points as the dashed rectangle in Fig 7A.

The increased fibril D-period in the calcified plate is surprising, when considering the well-established fact [28, 48] that the D-period for mineralized collagen in bone (for example) is consistently lower than hydrated unmineralized collagen. Mineralisation inside and around the fibrils has been linked to dehydration of collagen [26, 27], which leads to a lower D-period (~65 nm). The reasons for the reduction could include axial shrinkage [26] or a change in molecular tilt [24]. However, biochemical differences between Type I (bone) and II collagen (calcified cartilage) may couple with altered intra- and interfibrillar mineralisation patterns in

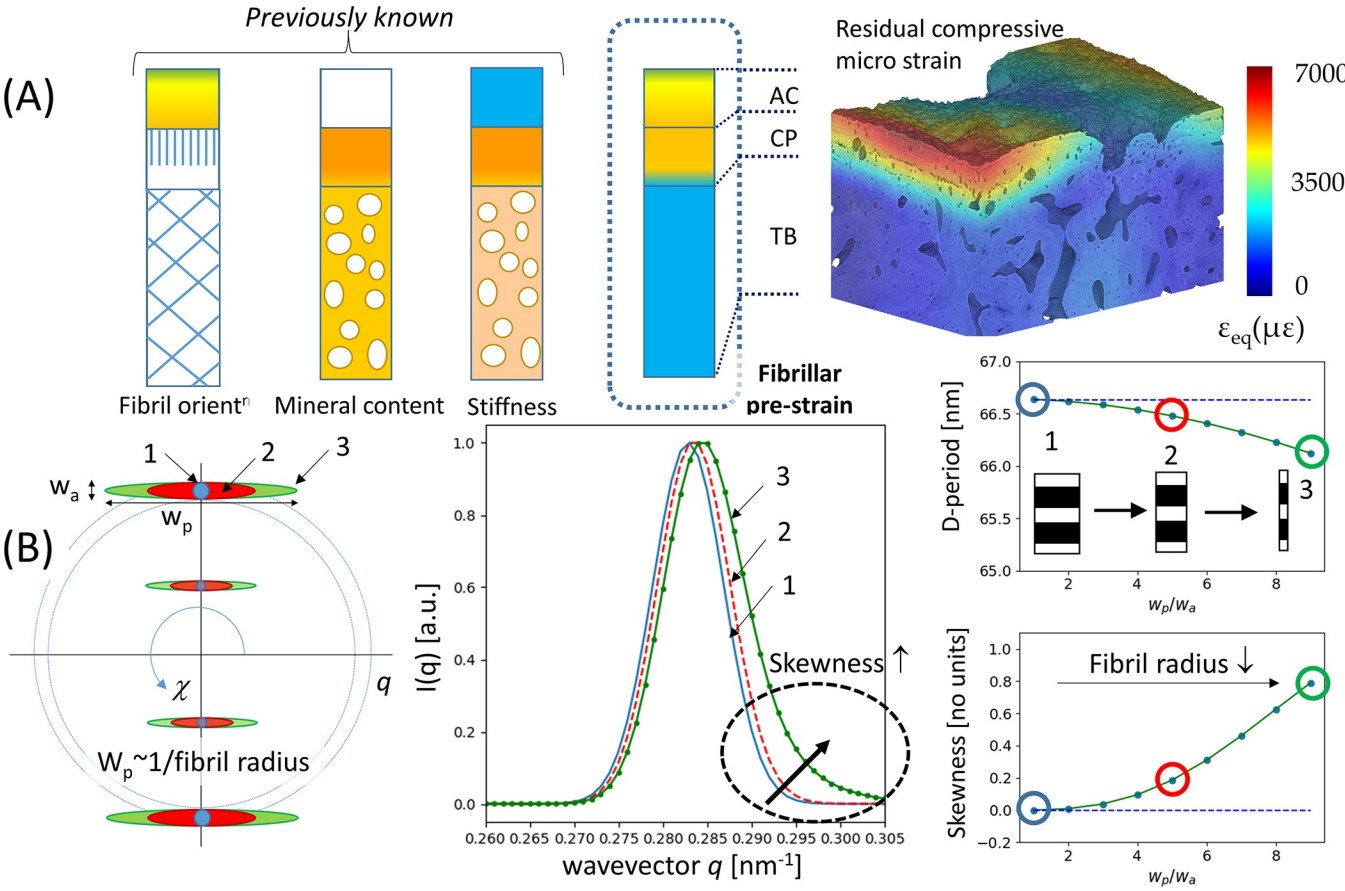

**Fig 9. Schematic overview and X-ray fibrillar model.** (A) Correlation between nanoscale fibrillar pre-strain and microscale residual compressive strain. Left: schematic of fibrillar pre-strain gradients (this work; Figs 2, 3 and 6) across the AC, CP, and TB in the bovine MCP joint. Right: From independent work on the medial condyle of bovine tibiae [49], residual compressive strain (after loading) visualised using microcomputed tomography (CT) digital volume correlation (DVC). The calcified plate (both CC and SCB) are seen as regions with both high fibrillar tensile pre-strain (in unloaded state) and high compressive strain (energy storage) after loading. (B) Fibre-diffraction simulations showing how increasing fibril radius affects the skewness of the peak and estimated D-period. *Left*: schematic of meridional SAXS ellipsoidal peaks from a single fibril for different ratios of parallel to axial peak widths $w_p/w_a$. Blue: $w_p/w_a = 1$; red: $w_p/w_a = 5$; green: $w_p/w_a = 10$. *Middle*: Simulated I(q) plots for 3rd order meridional peak for these ratios (with D-period kept constant), showing increasing skewness as $w_p/w_a$ increases, along with a slight rightward peak shift. *Right*: effect of increasing $w_p/w_a$ from 1 to 10 on (top): D-period calculated by moments method and (bottom) peak skewness, showing that the moment method underestimates the D-period as fibril radius decreases (1→2→3).

calcified cartilage leading to a larger D-period. In contrast, the D-period of ~65 nm for trabecular bone is consistent with previous studies of Type I mineralized collagen [16]. It is noted that calling the changes in D-period as changes in pre-strain is an approximation, since it assumes that the fibrils have similar zero-stress D-periods (or reference states) across the tissue types (articular cartilage, calcified plate and trabecular bone). Further, the change in pre-strain may be associated with alterations in molecular level tilt [22, 24] or ordering of tropocollagen molecules in the fibril [18], which are intrafibrillar mechanisms we do not consider here.

The higher resolution of the SAXS configuration compared to previous configurations [9, 18] means some features like the peak asymmetry are visible where before they were hidden by beam divergence. We believe the finite fibre diameter is influencing this asymmetry via a fibre-diffraction effect [50], conceptually sketched in Fig 9B. Fibrils with axial electron-density periodicity of D and radii R exhibit a set of parallel ellipsoidal layer-line reflections (spacing $2\pi/D$), with finite width ($w_p$: p: parallel) proportional to 1/R perpendicular to the fibril axis [50]. Thinner fibrils will have greater $w_p$ (e.g. case 3 (thin fibril) vs case 1 (thick fibril) in Fig 9B); the

axial width $w_a$ is kept constant in this sketch. The polar azimuthal (circular) average of flat ellipsoids parallel to the $q_x$ axis thus leads to a rightward skew asymmetry of the integrated peak (simulated data shown in Fig 9B, centre), as the tails of the ellipsoids contribute to larger wavevector values in the azimuthal ($\chi$) integration. As a result, the measured D-period (via weighted moments) underestimates the true D-period (as $q_3 = 6\pi/D$), with an increasing skew correlated to a reduced measured D-period. Fig 5B shows that (especially in cartilage) there is pronounced rightward skew of the I(q) peak. As a result, the estimated D-period via our non-parametric method is likely an underestimate in these regions, and the pre-strain level in both the AC- and CP-regions are even larger than estimated here, and the differences with respect to the underlying trabecular bone maintained.

To confirm the above, we carried out a simulation-experimental test (Supplementary Information: S1 Appendix), where the skew-effect was incorporated. Briefly, using fibre-diffraction models of the type shown in Fig 9, model SAXS peak functions with varying levels of (fibre-diameter induced) skew were generated for a range of axial and parallel peak widths $w_a$ and $w_p$. By matching the experimentally observable skew and axial width parameters for a representative sample, the true D-period can be estimated (S6 Fig). It is seen that while the cartilaginous tissue D-period is elevated in comparison to the first-moment method, the difference between the high D-period in the cartilaginous tissue and the lower D-period in the underlying trabecular bone is maintained. Future work will include concurrently fitting the azimuthal and radial intensity profiles, to independently determine $w_p$ as well as D-period, thus enabling a point-wise estimation of this effect. For the current results, we note only that the difference between the D-period of the articular cartilage/calcified plate and the trabecular bone is maintained.

The finer-structure gradient at the bone-cartilage interface, and the colour-contrasts in Figs 6 and 7, are intriguing and we believe are linked to the two-phase calcified cartilage and cortical subchondral bone material comprising the calcified plate. The length scale of ~70 μm over which the D-period rises from a local minimum right at the DZ/CP interface is similar to previously reported thicknesses of the tidemark [10], and suggests a compliant, less pre-strained region right at the interface. The more homogeneous fibrillar alignment ($\rho$) in the upper half of the calcified plate in Fig 7A versus the more heterogeneous distribution further down suggests a division between calcified cartilage (toward the articular cartilage) and the subchondral bone (toward the trabecular bone). Assuming this assignment, our results demonstrate that the high degree of fibrillar alignment and uniform orientation in the calcified cartilage (as shown earlier [10, 51, 52]) is coupled with a rapid rise in fibril D-period which reaches ~65.8 nm (typical of uncalcified cartilage [9]) and remains constant for ~50 μm in the calcified cartilage and a further 50 microns in the subchondral bone, followed by a decrease over ~100 μm in the subchondral bone down to typical ~65 nm values in the trabecular bone.

We speculate that the increased tensile fibril-level pre-strain in the calcified plate (Figs 2 and 3) has a biomechanical function related to increased elastic energy storage under compression. In the calcified plate, stiff fibrils are oriented parallel to the compressive loading direction (as shown earlier, e.g. [10]) and in a state of tensile pre-strain relative to the fibrils in trabecular bone. On mechanical loading, under compression, the pre-stretched fibrils can compress down from the pre-strain level down to zero, before compressive failure mechanisms like buckling start becoming operative. This mechanism is supported by prior work on tibial joints showing large residual strains in the calcified interface measured by digital volume correlation [49], and is shown schematically in Fig 9. This schematic Figure shows how the zones of high nanoscale fibrillar pre-strain (from this work) in the calcified plate correlate closely with the high microscale residual compressive strains at the bone-cartilage interface after compressive loading (measured with DVC methods on tomographic (CT) images [49]). A limitation of this

comparison is that the joint analysed in [49] is the bovine tibia, whereas here we have tested the bovine MCP joint. Nevertheless, it can be speculated that this natural gradient in ultra-structural tissue properties may enable absorption (and release) of significant stored elastic energy in regular biomechanical loading, with a material energy density of $\frac{1}{2} E \epsilon^2$ where E is the tissue elastic modulus of E~10–20 GPa [49], and $\epsilon$ the fibril pre-strain level of ~1.5%. Indeed, disruption of this ECM material-level energy absorption mechanism due to tidemark duplication in OA could be a biomarker-level indicator of OA progression.

Turning to the limitations of our work, a basic technical drawback is the use of 2D SAXS mapping of what is intrinsically a 3D cylindrical tissue object. Nevertheless, we mitigate these effects by choosing a cylindrical sample geometry, with relatively tissue homogeneity along the X-ray beam direction, coupled with a limited lateral width across the scan versus the height (high aspect ratio). Further, the variations we are interested in are along the length of the sample, rather than through the thickness, and are thus relatively insensitive to such 3D effects. For full-scale 3D reconstruction of the depth-dependent nanostructure, SAXS tensor tomography [36, 53] will be essential. Secondly, we limit ourselves to static scanning (no in situ mechanical loading) and leave spatial mapping of fibrillar mechanics under loading to future work. Thirdly, in terms of potential artefacts induced by sample preparation and experimentation, frictional heating during the coring process could damage the tissue and change the structure, and to prevent this, we use a slow coring under constant saline irrigation and cooling of the sample. To prevent any dehydration of the sample during the scanning (which will change the structure), the sample is kept fully immersed in physiological saline. Further, to minimize ultrastructural damage to the fibril structure due to X-ray radiation, in our protocol each tissue location on the sample is exposed only once to the beam (1.0s/point). Radiation-damage tests by repeated 1.0s SAXS exposures of the same cartilage tissue point show reductions of meridional peak intensity (characteristic of radiation damage) only after 4.0s of exposure, indicating that the single exposure scans are too short to cause damage. Fourthly, we have limited our sampling to a specific condylar anatomical site in the MCP joint and given that *in-vivo* strains vary across the joint, it would be interesting to test if nanomechanical parameters like pre-strain shows a correlation with varying physiological force levels across the joint. Lastly, we have not yet considered changes in OA and ageing, to establish a baseline characterisation of the normal bone-cartilage interface in an animal model; a recent study [11] has shown that nanoscale mineral particle thickness is significantly different in OA versus normal human patients and investigating in situ mechanical response in aged human cartilage would no doubt be of importance.

The use of scalar measures like SAXS intensity along with orientational information like fibrillar alignment ρ and direction $\chi_0$ enables us to classify tissue location based on properties of the oriented collagenous ECM (as shown in Fig 2A). The method allows discrimination within the AC (superficial, transitional, and deep zones) as well as between the AC, CP/SCB and TB regions. More advanced machine-learning classification methods have been used on the diffuse SAXS signal enabling distinction of the CP and SCB as well [11]; it would be of interest to apply such methods to the current data in future analysis. The nonparametric method for peak estimation arises from the pronounced peak asymmetry and the likely limits of linear background interpolation without a model. Recent analysis on pericardium collagen using step-function scattering models may, similarly, point the way to a the better analytical fitting of the peak profiles [54]. Nevertheless, the clear distinction between the meridional peaks (particularly the peak positions of the TB viz the CP/SCB region) is visible in the raw data itself (Fig 5) and will likely not be affected by such further analyses.

In summary, we have demonstrated the existence of a multiscale gradient in the fibrillar pre-strain across the bone-cartilage interface in bovine joints, along with an elevated pre-strain

mainly in the CC zone. The elevated pre-strain in the CP contrasts with the lower values in the underlying TB. Investigating the link between this nanoscale ECM structural change and the in-vivo biological growth dynamics would be interesting. Further, given that the bone-cartilage interface is subjected to considerable shear strains during loading [14], and disruptions to the interface have been implicated in OA progression [55, 56], understanding how the ECM architecture of the interface is designed to resist physiological loading, and how it may be altered in injury, ageing and musculoskeletal degeneration would be worth investigating to both understand the process and as well as to define better structural biomarkers of joint degeneration.

## Supporting information

**S1 Fig. Region classification of samples.** Representation of the region classification into SZ, TZ, DZ, CP and TB, for the 6 samples used in the analysis. Note that Sample 2 does not have an observable superficial (SZ) zone. The colour indicates the type of tissue region across BCU.
(TIF)

**S2 Fig. Histograms of SAXS parameters.** *Top to bottom*: Histograms of the D-period, fibril orientation and $\rho$ for the representative sample in Figs 2 and 3, for (left to right) the SZ, TZ, DZ, CP and TB tissue regions, respectively. The smooth lines are the associated kernel density distribution estimates and are shown for visualisation only.
(TIF)

**S3 Fig. SAXS scans of full-length scan.** Colour map of the depth-wise variation in SAXS derived parameters across bovine bone-cartilage core of 5mm length and 2mm diameter for a full-length scan. This single sample was scanned across a greater depth in the trabecular bone but is otherwise similar to the samples imaged in Figs 2 and 3 in the main text. Step size of 20 microns, sample size ~0.38mm width, 5mm length. In this sample we were unable to resolve the thin superficial zone (SZ) at the top. Colour plots display: (A) Regions TZ: transitional zone, DZ: deep zone, CP: calcified plate, and TB: trabecular bone (as in Fig 1A, main text), (B) D-period (nm), reflecting collagen pre-strain, (C) total SAXS intensity (a.u.); areas of high intensity correspond to mineral-dense regions, (D) Total SAXS intensity from the background-corrected meridional collagen peak intensity; here, high intensity is observed in articular cartilage and (E) degree of orientation $\rho$ (a.u.), showing high values in the deep zone (DZ), intermediate values in the calcified plate (CP) and transitional zone (TZ), and low values in trabecular bone (TB).
(TIF)

**S4 Fig. Asymmetry in I(q) plots.** (A) Azimuthally integrated I(q) plots from the different tissue zones (SZ, TZ, DZ, CP and TB), corrected for diffuse background. I(q) plots are laterally averaged across sample width at specific vertical depths from cartilage surface; x-axis wavevector $q$ is in nm$^{-1}$ (B) Plots from (A) normalized to maximum peak intensity, to show the peak shape variations more clearly.
(TIF)

**S5 Fig. Fibre diffraction modelling A.** (A) Solid lines denote the simulated increase in skew as the equatorial width $w_p$ increases; as $w_a$ is different for each tissue zone, the curves are separate; $w_p$ is in units of nm$^{-1}$. Filled circles indicate the experimentally determined skew (from S4 Fig) for each zone. (B) The predicted increase in peak position $q_0$ with skew (in nm$^{-1}$), as demonstrated in Fig 9 (main text); solid circles denote the expected increase for different tissue zones, using skew values obtained in (A).
(TIF)

**S6 Fig. Fibre diffraction modelling B.** Blue bars: D-period (in nm) for the different I(q) curves in S4 Fig, calculated using the first moment of area method used in the main text. Black bars: True D-period, obtained by correcting for the artificial increase in D-period due to the skew, demonstrated in S5B Fig.
(TIF)

**S1 Table. Results from statistical testing of each nanoscale parameter for differences in regions across the BCU for full-length scan.** ANOVA tests followed by TUKEY HSD tests were carried out on the data. P values indicate statistical significance, where *** is (p<0.001), ** is (p<0.01), * is (p<0.05) and ns is non-significant difference between regions.
(PDF)

**S2 Table. Results from statistical testing of each nanoscale parameter for differences across tissue regions in the BCU.** Data from the 6 regular scan samples. 1-way ANOVA test (with tissue zone as the factor) was carried out on the data, with each scan point labelled as belonging to a specific tissue zone. P values indicate statistical significance, where *** is (p<0.001), ** is (p<0.01), * is (p<0.05) and ns is non-significant difference between regions.
(PDF)

**S1 Appendix. Description of the fibre diffraction modelling procedure.**
(PDF)

## Acknowledgments

We thank Andy Smith and Olga Shebanova (Diamond Light Source (DLS)) for excellent technical support during the synchrotron beamtime.

## Author Contributions

**Conceptualization:** Waqas Badar, Martin M. Knight, Himadri S. Gupta.

**Data curation:** Waqas Badar.

**Formal analysis:** Waqas Badar, Husna Ali, Olivia N. Brooker, Elis Newham, Peter Fratzl, Himadri S. Gupta.

**Funding acquisition:** Martin M. Knight, Himadri S. Gupta.

**Investigation:** Waqas Badar, Husna Ali, Olivia N. Brooker, Elis Newham, Tim Snow, Nicholas J. Terrill, Gianluca Tozzi, Himadri S. Gupta.

**Methodology:** Waqas Badar, Husna Ali, Olivia N. Brooker, Elis Newham, Peter Fratzl, Martin M. Knight, Himadri S. Gupta.

**Project administration:** Waqas Badar, Martin M. Knight, Himadri S. Gupta.

**Resources:** Tim Snow, Nicholas J. Terrill, Gianluca Tozzi.

**Software:** Waqas Badar, Husna Ali, Olivia N. Brooker, Elis Newham, Tim Snow, Nicholas J. Terrill, Himadri S. Gupta.

**Supervision:** Nicholas J. Terrill, Martin M. Knight, Himadri S. Gupta.

**Visualization:** Waqas Badar, Husna Ali, Olivia N. Brooker, Elis Newham, Gianluca Tozzi, Himadri S. Gupta.

**Writing – original draft:** Waqas Badar, Himadri S. Gupta.

**Writing – review & editing:** Waqas Badar, Husna Ali, Olivia N. Brooker, Elis Newham, Tim Snow, Nicholas J. Terrill, Gianluca Tozzi, Peter Fratzl, Martin M. Knight, Himadri S. Gupta.

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
