## [Decision Letter · Decision Letter 0]

17 Jun 2022

PONE-D-22-11854Collagen Pre-Strain Discontinuity at the Bone-Cartilage InterfacePLOS ONE

Dear Dr. Gupta,

Thank you for submitting your manuscript to PLOS ONE. After careful consideration, we feel that it has merit but does not fully meet PLOS ONE’s publication criteria as it currently stands. Therefore, we invite you to submit a revised version of the manuscript that addresses the points raised during the review process.

Please, address all the minor comments addressed by both reviewers.

We look forward to receiving your revised manuscript.

Kind regards,

Antonio Riveiro Rodríguez, PhD

Academic Editor

PLOS ONE

Journal Requirements:

2. Please provide in the Methods section the name and location of the slaughterhouse from which samples were obtained.

[HSG (BB/R003610/1, EP/V011235/1) and NJT (BB/R003610/1, EP/V011235/1) thank the BBSRC and EPSRC for research grant funding. HSG thanks MRC (MR/R025673/1) for funding. We thank Diamond Light Source (DLS) for the generous award of synchrotron beamtime (SM25602-2).]

 [HSG: BB/R003610/1, EP/V011235/1, MR/R025673/1, SM25602-2

NJT: BB/R003610/1, EP/V011235/1, SM25602-2

BB/R003610/1: Biotechnology and Biological Sciences Research Council (https://www.ukri.org/councils/bbsrc/)

EP/V011235/1: Engineering and Physical Sciences Research Council (EPSRC) (https://www.ukri.org/councils/epsrc/)

MR/R025673/1: Medical Research Council 

(https://www.ukri.org/councils/epsrc/)

SM25602-2: Diamond Light Source (STFC) (www.diamond.ac.uk)

The funders had no role in study design, data collection and analysis, decision to publish, or preparation of the manuscript.]

Reviewers' comments:

Reviewer's Responses to Questions

**Comments to the Author**

1. Is the manuscript technically sound, and do the data support the conclusions?

Reviewer #1: Yes

Reviewer #2: Yes

2. Has the statistical analysis been performed appropriately and rigorously? 

Reviewer #1: Yes

Reviewer #2: Yes

3. Have the authors made all data underlying the findings in their manuscript fully available?

Reviewer #1: Yes

Reviewer #2: Yes

4. Is the manuscript presented in an intelligible fashion and written in standard English?

Reviewer #1: Yes

Reviewer #2: Yes

5. Review Comments to the Author

Reviewer #1: This paper, from an outstanding team of world-class researchers, reports an exciting new result on the bone-cartilage unit. The collagen fibril appears to possess an internal tensile pre-strain gradient. Such a discovery can only be possible thanks to X-ray scattering, a technique in which the authors have a track record.

Indeed, the magnitude of such pre-strain (0.6%) is so tiny that only SAXS can detect it.

The methods are sound, and results carefully elaborated. I recommend acceptance.

Minor comment: The acronym AC is used early in the manuscript but, if I'm not mistaken, is defined only for the first time on line 293.

Reviewer #2: 1. Please check line 147 " ... using a high high-speed saw. "

2. Please improve the quality of Figures. The text in Figure 1 is difficult to recognize.

3. Please explain how the sample preparation method and the state of the sample during X-ray measuring might affect the experimental results? How to avoid these problems in this work?

6. PLOS authors have the option to publish the peer review history of their article (what does this mean?). If published, this will include your full peer review and any attached files.

Reviewer #1: No

Reviewer #2: No

---

## [Author Response · Author response to Decision Letter 0]

9 Aug 2022

Please see our response to reviewers which is uploaded as a separate file.

---

## [Decision Letter · Decision Letter 1]

17 Aug 2022

Collagen Pre-Strain Discontinuity at the Bone-Cartilage Interface

PONE-D-22-11854R1

Dear Dr. Gupta,

We’re pleased to inform you that your manuscript has been judged scientifically suitable for publication and will be formally accepted for publication once it meets all outstanding technical requirements.

Kind regards,

Antonio Riveiro Rodríguez, PhD

Academic Editor

PLOS ONE

Reviewers' comments:

Reviewer's Responses to Questions

**Comments to the Author**

1. If the authors have adequately addressed your comments raised in a previous round of review and you feel that this manuscript is now acceptable for publication, you may indicate that here to bypass the “Comments to the Author” section, enter your conflict of interest statement in the “Confidential to Editor” section, and submit your "Accept" recommendation.

Reviewer #1: All comments have been addressed

Reviewer #2: All comments have been addressed

2. Is the manuscript technically sound, and do the data support the conclusions?

Reviewer #1: Yes

Reviewer #2: Yes

3. Has the statistical analysis been performed appropriately and rigorously? 

Reviewer #1: Yes

Reviewer #2: Yes

4. Have the authors made all data underlying the findings in their manuscript fully available?

Reviewer #1: Yes

Reviewer #2: Yes

5. Is the manuscript presented in an intelligible fashion and written in standard English?

Reviewer #1: Yes

Reviewer #2: Yes

6. Review Comments to the Author

Reviewer #1: (No Response)

Reviewer #2: (No Response)

7. PLOS authors have the option to publish the peer review history of their article (what does this mean?). If published, this will include your full peer review and any attached files.

Reviewer #1: No

Reviewer #2: No

---

## [Editor Report · Acceptance letter]

2 Sep 2022

PONE-D-22-11854R1 

Collagen Pre-strain Discontinuity at the Bone - Cartilage Interface 

Dear Dr. Gupta:

I'm pleased to inform you that your manuscript has been deemed suitable for publication in PLOS ONE. Congratulations! Your manuscript is now with our production department. 

Kind regards, 

on behalf of

Dr. Antonio Riveiro Rodríguez 

Academic Editor

PLOS ONE